# Phytochemical Analysis of *Acacia*
*ehrenbergiana* (Hayne) Grown in Qatar: Identification of Active Ingredients and Their Biological Activities

**DOI:** 10.3390/molecules27196400

**Published:** 2022-09-28

**Authors:** Vandana Thotathil, Hanan Hosni Rizk, Ameena Fakrooh, Lakshmaiah Sreerama

**Affiliations:** Department of Chemistry and Earth Sciences, Chemistry Program, College of Arts and Sciences, Qatar University, Doha P.O. Box 2713, Qatar

**Keywords:** *Acacia ehrenbergiana*, phytochemical analysis, antibacterial activities, antioxidant activities, Salam

## Abstract

*Acacia ehrenbergiana* (Hayne), also known as Salam, is a highly drought resistant shrub distributed in North and East Africa, and the Arabian Peninsula. The plant is gathered for its gum and fiber, and is an important legume species for indigenous populations. In this study, the phytochemical analysis, antibacterial, and antioxidant properties of various alcoholic and aqueous extracts of *Acacia ehrenbergiana* grown in Qatar were investigated. The qualitative phytochemical screening of this species exhibited the presence of glycosides, tannins, flavonoids, terpenoids, saponins, phenol, and anthraquinones in various extracts. The agar diffusion method was performed to check the antibacterial activity. The acetone and ethanol extracts showed 85% antibacterial activity of the control against Gram-negative *E. coli*, while the acetone extract had 65% activity against the *Bacillus* Gram-positive species. The highest activity against *Staphylococcus aureus* was 65% for the butanol extract. The antioxidant capacities were evaluated by the DPPH method. Various extracts exhibited antioxidant activities similar to or higher than standard antioxidants, with the highest percent inhibition of 95% for the acetone and ethanol extracts. The acetone extracts were further purified by reverse phase combiflash chromatography followed by HPLC. Three of the pure compounds isolated were subjected to MS, FTIR, and NMR spectral analysis and were found to be stigmasterol, spinasterol, and theogallin. In conclusion, the observed antibacterial and antioxidant activities as well as the presence of secondary metabolites with potential medicinal activities makes *Acacia ehrenbergiana* a potent valuable endemic medicinal plant.

## 1. Introduction

From time immemorial, even before the dawn of modern medicine, man has resorted to nature for remedies against various diseases and ailments [1,2]. Due to their structural and chemical diversity, plants are one of the most important natural sources of therapeutic agents and bioactive compounds. Traditionally, plant extracts or concoctions are used to treat diseases without the isolation of the active ingredient. Modern medicine requires the active ingredient to be isolated. Active ingredients obtained from medicinal plants play a vital role in modern drug industries and drug design.

In the small Arabian Peninsula of Qatar, natural vegetation is limited and restricted to specific landforms. Even though there are some studies available on the isolation and structural elucidation of compounds from Qatari plants, data on their antibacterial activities are scarce [3]. Among the very few trees present in the region, *Acacia ehrenbergiana* (Hayne), known as Salam in Arabic, is used by locals for shade, firewood, and fodder for camels [4,5]. It belongs to the genus Acacia of the Leguminosae (or Fabaceae) family. It is also one of the most drought-tolerant plants among the common African acacias grown in the rainfall belts (50–400 mm). Apart from Qatar, it is found in the Sahara, northern Sahel, East Africa, and the Arabian Peninsula. In the latter regions, *Acacia ehrenbergiana* is commonly used as feed for livestock. The genus Acacia, which is rich in polyphenols, is found to exhibit molluscicidal activity and tanning properties [6]. The bark of all Acacia species contains varying quantities of tannins that are known to have astringent properties. Accordingly, they are used to treat diarrhea, dysentery, internal bleeding, and various skin conditions. The bark extracts are often used to wash wound and injuries and for wound healing [7,8]. Some of its preparations are used to treat hemorrhoids, perspiring feet, and some eye conditions [7,8]. A gum obtained from the plant is used as an emollient in ointments and shampoo [9]. In Yemen, Acacia trees are used in apiculture to produce the highest quality honey, rich in phenolic content and antioxidant properties [10]. Acacia trees are also an economic resource in Middle-Eastern Countries such as Yemen and Egypt to produce charcoal, firewood, and tar. Tar is a fluid extracted from charcoal and used for treating animal skin diseases [11].

It has been reported that the *Acacia ehrenbergiana* plant extract shows antibacterial activities [11,12,13], and some of the identified bioactive constituents include gallic acid, rutin 2″-O-α-L-rhamnopyranosyl, myricetin 3-O-β-D-rutinoside, rutin, myricetin 3-O-β-D-glucoside, quercetin 3-O-β-D-glucoside (isoquercitrin), myricetin, quercetin, and catechin [14]. A study on the ethanolic extract of *Acacia ehrenbergiana* showed a significant anti-inflammatory effect at a 300 mg/kg dose [15]. Another study conducted on the ethanol extract of the flower of *Acacia ehrenbergiana* identified 15 phytoconstituents with potent antibacterial activity against several bacterial strains including *S. aureus, E. fecalis, E. coli, P. mirabilis, P. aeruginosa,* and *K. pneumonia* [16].

Depending on the region where the plant is grown, the climatic conditions, the soil types and availability of moisture content in the soil, the secondary metabolites (phytochemicals) produced by the plants vary. Acacia grown in arid and dry conditions are of particular importance as their phytochemical constituents have not been not analyzed in detail [17].

With the continuous emergence of drug resistance in microbial strains, the search for new anti-microbial agents have been given high priority by natural product chemists and ethno-pharmacologists. In recent decades, investigations are ongoing for the evaluation of antioxidant activities of phytochemicals such as phenols, flavonoids, and tannins in plants. They have received much attention because of their potential role in the prevention of cellular damage, a common pathway for the development of cancers, aging, and a variety of other diseases. The aim of the current study was to report on the isolation and structural elucidation of compounds present in the extracts of the plant *Acacia ehrenbergiana* grown in the arid and hot climatic conditions of Qatar as well as to evaluate their antibacterial and antioxidant properties. We will also discuss the compounds identified in the acetone extracts and the role played by them in enhancing the antibacterial and antioxidant activities of the extracts of *Acacia ehrenbergiana.*

## 2. Results

### 2.1. Phytochemical Analysis of Acacia ehrenbergiana Grown in Qatar

The qualitative analysis of important phytochemicals in various extracts of *Acacia ehrenbergiana* grown in Qatar showed the presence of a variety of compounds (Table 1). Glycosides were found in all extracts except for the methanol extract. Tannins were present in all except for the acetone extract. Tests for flavonoids, terpenoids, phenol, and anthraquinones indicated their presence in all extracts. An interesting observation is that the anthraquinone test resulted in the formation of a greenish-yellow precipitate in the methanolic extracts rather than aa pink-violet or red coloration. Additionally, saponins were present in the ethanol extract.

### 2.2. Antibacterial Activity of Acacia ehrenbergiana Extracts

The disc diffusion method was used to assess the antibacterial activity of the extracts [21]. Extracts were tested at 5, 10, 25, 50, 100, and 200 µg/mL versus ampicillin 100 µg/mL as the control and the areas of inhibition were estimated in mm^2^ compared to the positive controls. The activities were dependent on the concentration of the extract, with the highest activity for 200 µg/mL (Table 2). The antibacterial activity was determined using Gram-negative *E. coli,* and Gram-positive *Bacillus* and *Staphylococcus aureus* species. The acetone and ethanol extracts displayed 85% antibacterial activity of the control against Gram-negative *E. coli*, while the acetone extract had 65% activity against the *Bacillus* Gram-positive species. The highest activity against *Staphylococcus aureus* was 65% for the butanol extract.

A comparison of the antibacterial activity of *Acacia ehrenbergiana* grown in different regions as reported in previous studies is shown in Table 3. As depicted, studies on *Acacia ehrenbergiana* grown in Sudan, Egypt, and Saudi Arabia showed a stronger activity against the Gram-positive strains for the ethanol and methanol extracts. In Qatar, the activity was higher against *E*. *coli* for the acetone extracts.

### 2.3. Antioxidant Activity of Acacia ehrenbergiana

The free radical scavenging activity of the investigated extracts was assessed using the direct DPPH assay [16]. In this assay, DPPH (deep purple colored free radical) turns into pale yellow when reduced by antioxidants present in the sample. Various extracts displayed different antioxidant activities measured as a percent inhibition with reference to known standard antioxidants, α tocopherol, and ascorbic acid (Figure 1). For example, the acetone and ethanol extracts (undiluted) showed a 95% antioxidant activity. This was higher than in both standards. When diluted 100-fold, both extracts retained antioxidant activities higher than the standards. The antioxidant activity was found to increase in the following order: methanol, ethyl acetate, butanol, acetone, and ethanol.

### 2.4. Characterization of Secondary Metabolites Present in Acetone Extracts of Acacia ehrenbergiana

The acetone extracts (showing higher antibacterial and antioxidant activity) were further subjected to reverse phase combiflash chromatography. The latter resulted in the collection of 10 fractions. Nine of these fractions contained secondary metabolites and were saved for the further identification of compounds. The purity of the above fractions was analyzed by TLC. Two of the nine fractions appeared to be pure compounds, as assessed by TLC analysis. All nine fractions were subjected to further purification by HPLC to isolate the pure compounds. This process led to the fractionation and isolation of at least 12 compounds. Of these 12 compounds, three compounds were isolated in large enough quantity so that they could be further characterized and identified by spectroscopy techniques (MS, FTIR, and NMR). We are in the process of isolating larger amounts of nine other compounds for their identification in the future. Fractionation and isolation of the biologically active compounds in other extracts of *Acacia ehrenbergiana* is ongoing. The three compounds isolated in larger quantities were identified based on the spectral characteristics and library searches to be stigmasterol, spinasterol, and the quinic acid derivative theogallin (Figure 2).

### 2.5. Antibacterial Activity of Isolated Compounds

The antibacterial action of the stigmasterol, spinasterol, and theogallin isolated compounds at 2000, 1000 and 500 µg/mL were evaluated as compared to the control chloramphenicol 200 µg/mL and mean diameter inhibition zones (M.D.I.Z) are shown in Table 4. Moreover, the antifungal activity was compared to fluconazole 1250 µg/mL as the control, as shown in Table 5.

## 3. Materials and Methods

### 3.1. Materials

DPPH (2,2-diphenyl-1-picrylhydrazyl) and organic solvents (acetone, ethanol, ethyl acetate, butanol, chloroform, and methanol) were purchased from Sigma-Aldrich Chemical Co., St. Louis, MO, USA. All other chemicals were available locally and were of analytical grade.

### 3.2. Methods

#### 3.2.1. Extraction of Phytochemicals and Fractionation

The aerial parts of *Acacia ehrenbergiana* plant, that is, twigs with bark, leaves, and flowers were collected from its natural habitat in the desert near Doha, Qatar. The plant material, twigs with bark, was ground into a fine powder, sieved (partial size <0.2 mm) and stored in sealed plastic bags in a clean and dry cabinet. About 50 g of the preparation was treated with various organic solvents and the soluble compounds were extracted by the Soxhlet solvent extraction method. The solvents used for this method were **a**: acetone, **b**: butanol, **c**: ethanol, **d**: ethyl acetate, **e**: methanol. The extracts were concentrated to approximately 10 mg/mL using a Rotovap (Rotavapor R210, Buchi & Flawil, Flawil, Switzerland) and stored at −20 °C until used further.

#### 3.2.2. Phytochemical Analysis

The extracts obtained were subjected to various functional group tests to identify the types of compounds present in them, as described previously [18,19,20]. The extracts that showed the highest levels of phytochemicals were further used for the isolation of the compounds.

#### 3.2.3. Antibacterial Activity Tests

In this study, the disc diffusion method was used to determine the antibacterial activities. Ampicillin 100 µg/mL was used as the positive control. Bacterial stock cultures (1-day old; with an absorbance of 1.2 at 600 nm) were used for these experiments. A small amount of stock culture (25 µL), enough to cover the plate, was spread uniformly with the aid of a bacteria cell spreader and left for 15 min so that the culture soaked up properly. Each plate was divided into four quadrants and each quadrant had a wick of filter paper in the middle of it. The positive control and test compounds were added to the filter paper wicks. Additional filter paper wicks were used in order to prevent overflowing of the solution. The plates were then incubated at 37 °C for 1-day. After the incubation was complete, the plates were examined for clear zones around the filters [21]. Ampicillin (100 μg/mL) and/or chloramphenicol (200 µg/mL) was used as the positive control.

#### 3.2.4. DPPH In Vitro Assay

The initial concentration of the extracts were adjusted to 100 µg/mL and were tested for antioxidant activities using the DPPH (2,2-diphenyl-1-picrylhydrazyl) method at 1×, 10×, and 100× dilutions. They were then compared to the antioxidant activities of the positive control α-tocopherol and ascorbic acid. The extract was added to DPPH (1 mL, 0.1 mM, methanol) in a test tube, shaken vigorously by hand and allowed to stand at 27 °C in a dark place for 45 min. The positive control sample was prepared according to the same procedure without any extract. The absorbance of the tested samples were measured using a spectrophotometer at the wavelength 517 nm. The total antioxidant activity of the tested crude extract samples was determined as the inhibition percentage and was calculated by using the well-established formula (see below) [24]:

The DPPH scavenging effect (%) or percent inhibition:



$$\frac{\mathrm{Absorbance}\;\mathrm{of}\;\mathrm{control}-\mathrm{Absorbance}\;\mathrm{of}\;\mathrm{sample}}{\mathrm{Absorabance}\;\mathrm{of}\;\mathrm{control}}\times 100$$



#### 3.2.5. Identification and Characterization of the Active Compounds

The plant extracts/fractions that exhibited the highest antioxidant activity (acetone extracts) was subjected to preparative HPTLC and HPLC to isolate various components. Various fractions recovered were tested for antioxidant activities and further purified by HPLC if necessary. HPTLC was performed on CAMAG^®^ HPTLC scanner III using hexane and ethyl acetate as the solvent in the ratio 7:3. HPLC was conducted with the aid of a Shimadzu SPD 20A HPLC-MS fitted with a C_18_ column (4.6 × 250 mm; 0.5 micron) with acetonitrile and water (15:85) with 0.1% phosphoric acid as the solvent at a flow rate of 1 mL/min. The UV spectra were recorded on a PDA detector. Pure compounds were characterized by various spectroscopic techniques (MS, FTIR, and NMR spectroscopy; See Appendix A). NMR spectra was recorded using a Bruker 400 MHz spectrometer for ^1^H and ^13^C using deuterated chloroform (CDCl_3_). Chemical shifts are expressed in δ (ppm) down-field from tetramethylsilane as an internal standard. The IR spectra were recorded with the aid of a Bruker FT-IR spectrophotometer from wavelengths of 4000–500 cm^−1^.

## 4. Discussion

Natural products represent a huge reservoir of bioactive compounds that have become essential components of the drug discovery process. The identification of secondary metabolites in plants will help interpret the therapeutic actions as well as help in designing chemically pure medications via the complete synthesis or partial modification of the identified compounds. In this study, *Acacia ehrenbergiana,* a shrub grown in Qatar in its natural desert environment with resistance to drought, was investigated for the presence of phytochemicals, antibacterial, and antioxidant properties as well as isolating and characterizing the most active ingredients in some of the plant extracts.

Screening experiments showed the presence of different phytochemicals such as tannins, flavonoids, terpenoids, saponins, phenol, and anthraquinones in the *Acacia ehrenbergiana* extracts. These phytochemicals have been shown to have extensive biological and therapeutic properties. For example, flavonoids are known for their antioxidant properties, terpenoids are reported to have anti-inflammatory activity, and saponins have antibacterial, anti-inflammatory, immune boosting, and cholesterol reducing medical effects whereas phenols possess antioxidant and anticancer effects [25]. Anthraquinones are used for their laxative effects. In this study, *Acacia ehrenbergiana* was shown to have a unique greenish-yellow rather than the expected pink-violet coloration. This suggests that it may contain new or modified anthraquinone compounds.

There are earlier studies establishing a wide range of antibacterial activities for *Acacia ehrenbergiana* [4,12,13,22]. On testing the antibacterial activities of *Acacia ehrenbergiana,* a significant effect was found for the acetone and ethanol extracts on Gram-negative *E. coli*. A study conducted in Sudan stated that Gram-positive *Bacillus subtilis* and *Staphylococcus aureus* were the most susceptible to the methanol and ethanolic extracts of leaves [12]. However, in our studies, we found that methanol extracts showed a weak antibacterial activity. The reason for this contrasting observation could be attributed to the difference in the concentration of the extract as well as the fact that we used twig and bark extracts. We found that the antibacterial activity was dependent on the concentration of the extract, and was consistent with the report of Rahim and associates [12]. There is evidence to show that depending on the place and climatic conditions, the presence of biologically active constituents may vary. Therefore, investigators have also proposed that the presence of varying concentrations of biologically active constituents such as alkaloids, tannins, saponins, and flavonoids might be responsible for the antibacterial activity, especially in the leaf and florets, which are rich in alkaloids and saponins. The latter may explain the traditional use of acacia leaves for the treatment of bacterial infections as well as wound healing capabilities [4,13]

The antioxidant activities measured by the DDPH assay show that the acetone and ethanol extracts exhibited 85% inhibition when diluted 100 times. On the other hand, α-tocopherol and ascorbic acid exhibited 78% and 85% inhibition, respectively, under similar conditions. Furthermore, various extracts of *Acacia ehrenbergiana* displayed antioxidant activities to varying extents. As depicted in various studies, the strong correlation between the content of phenols, saponins, flavonoids, tannins, and radical scavenging activity indicates that these phytochemical constituents are major contributors to the antioxidant properties [26]. All of the above compounds were also found in *Acacia ehrenbergiana.* It is noteworthy that *Acacia ehrenbergiana* had the highest antioxidant activity and total phenolic content among the other acacia species. Due to this, the best quality of honey was harvested from *Acacia ehrenbergiana* [10]. The stigmasterol and spinasterol isolated and identified in this study were phytosterols present in plant oils and are reported to have antioxidant properties [26], although some studies found them to be weak [27]. The high solubility of sterols observed in the ethyl acetate extract might contribute to the antioxidant activity observed in this study for this extract. This was similar to the standards, however, further studies of their antioxidant potential need to be explored. Therefore, this species can be potent target to derive drugs with antioxidant properties. However, more extensive studies using in vivo models are still needed to confirm this property.

Previous studies have shown that stigmasterol (100 μg/mL) inhibits the growth of various organisms with a mean zone of inhibition ranging from 23 mm to 30 mm. The activity of stigmasterol was much stronger compared to ciprofloxacin (5 μg/mL), the standard antibacterial drug, and fluconazole (5 μg/mL), the antifungal agent. The minimum inhibitory concentration (MIC) and the minimum bactericidal/fungicidal concentration (MBC/MFC) of stigmasterol ranges from 6.25 μg/mL to 25 μg/mL and from 12.5 μg/mL to 50 μg/mL, respectively [25]. However, in our study, stigmasterol exhibited much weaker inhibition zones at 500, 1000, and 2000 µg/mL; 2 mm antibacterial and 1 mm antifungal activities compared to the compounds tested. The other metabolites spinasterol and theogallin also showed similarly weak antibacterial and antifungal activities.

Stigmasterol exhibits a wide spectrum of pharmacological activities including cholesterol lowering, thyroxin lowering, hypoglycemic, anti-inflammatory, and anti-osteoarthritic activities [12]. This compound is also effective against various disease conditions including arthritis, diabetes, cardiovascular ailments, renal disorders, hepatic toxicity, microbial infections, and cancer. Stigmasterol inhibits several pro-inflammatory and matrix degradation mediators typically involved in osteoarthritis-induced cartilage degradation, at least in part through the inhibition of the NF-kappa B pathway [28]. Spinasterol, another phytosterol identified in this study, exhibits anticancer, antiproliferative, COX inhibition with antinociceptive effects, and anti-inflammatory actions [29,30]. The anti-inflammatory effects of the *Acacia ehrenbergiana* extracts could be partially attributed to the stigmasterol and spinasterol compounds. The ethanolic extracts of leaves were recently reported to have in vivo anti-inflammatory activity, almost similar to the anti-inflammatory effect of diclofenac sodium [15]. The presence of saponins and flavonoids, found in the ethanolic extracts of our study, might contribute to the anti-inflammatory properties exhibited by *Acacia ehrenbergiana* [31,32].

Theogallin, identified in the acetone extract in this study, is a polyphenolic quinic acid derivative that is characterized as a famous umami enhancing compound in tea [33]. Dempfel and his associates indicated that theogallin, or its metabolite quinic acid, possesses mild cognition enhancing effects in rats [34]. Tea polyphenols are known for their antioxidant properties, and this might partly explain the high antioxidant activity of the acetone extract in this study [35]. We did not find data evaluating the antibacterial activity of theogallin in the literature, and in this study, theogallin did not exhibit an antibacterial activity.

Given the discussion above, we strongly believe that the identification of various compounds in *Acacia ehrenbergiana,* a plant widely spread in Qatar and dry lands of other countries, would be of medicinal and economic value. From our studies, we conclude that *Acacia ehrenbergiana* is a potential source of phytochemicals and secondary metabolites with antibacterial, antioxidant, and possibly other pharmacological activities. Therefore, it can be a promising source of molecules of interest that can be used as natural drugs or in the design of their synthetic variations.

## Figures and Tables

**Figure 1 molecules-27-06400-f001:**
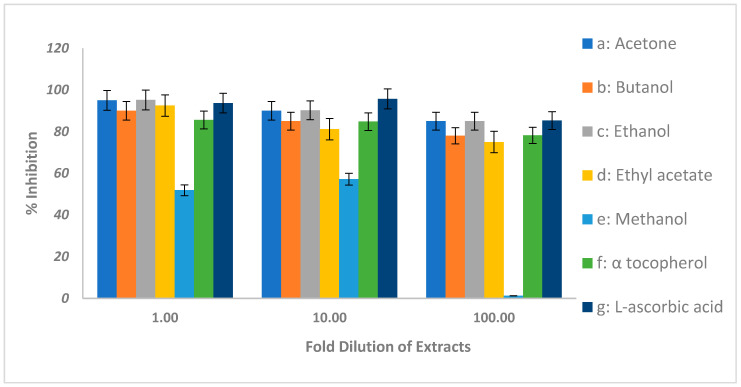
The antioxidant activity of the *Acacia ehrenbergiana* extracts by the DPPH method. Various extracts prepared from *Acacia ehrenbergiana* were subjected to the DPPH assay as described in the Materials and Methods section. The inhibition activity was compared to known antioxidants (α-tocopherol and ascorbic acid).

**Figure 2 molecules-27-06400-f002:**
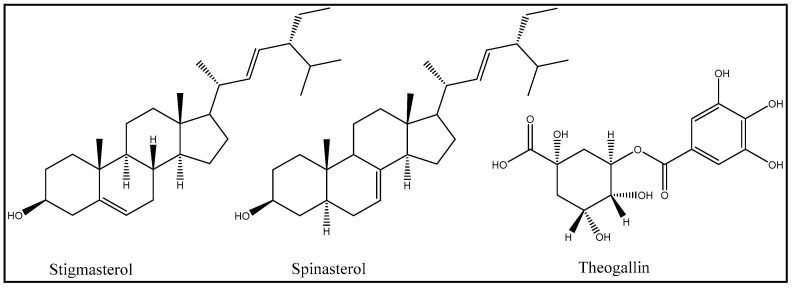
The major secondary metabolites identified in the acetone extracts of *Acacia ehrenbergiana*.

**Table 1 molecules-27-06400-t001:** The phytochemical analysis of *Acacia ehrenbergiana* grown in Qatar ^1^.

#	Test	Procedure	Expected Positive Test Result	a ^2^	b ^2^	c ^2^	d ^2^	e ^2^
1	Alkaloids	Wagner’s Test	Brown/Reddish precipitate	−	−	−	−	−
2	Glycosides	Kellar–Kiliani’s Test	Brown ring at the junction	+	+	+	+	−
3	Tannins	Braymer’s Test	Dark blue or greenish grey coloration of the solution	−	+	+	+	+
4	Flavonoids	Alkaline Reagent Test (A Drop of 10%NaOH)	Intense yellow color	+	+	+	+	+
5	Terpenoids	Salkowski Test	Reddish brown ring at the junction	+	+	+	+	+
6	Saponins	Foam Test	Stable froth produced	−	−	+	−	−
7	Phenols	FeCl_3_ Test	Blue-green	+	+	+	+	+
8	Anthraquinones	1 mL benzene and 3–5 drops of 26% NH_3_	Pink or violet or red coloration in the ammonical layer ^3^	+	+	+	+	+
9	Phlobatannins	2 mL extract was boiled with 2 mL of 1% hydrochloric acid HCl	Red ppt	−	−	−	−	−
10	Anthocyanins	2 mL HCl (2M) + NH_3_	Pinkish-red to bluish violet color	−	−	−	−	−
11	Proteins	Xanthoproteic Test	White precipitate	−	−	−	−	−

^1^ The detailed experimental procedures used for the phytochemical tests were described previously [18,19,20]. ^2^ a: acetone extract, b: butanol extract, c: ethanol extract, d: ethyl acetate extract, e: methanol extract. ^3^ The anthraquinones are expected to produce a pink-violet or red coloration in the ammonical layer, but a greenish-yellow precipitate formation was observed.

**Table 2 molecules-27-06400-t002:** The antibacterial activity of *Acacia ehrenbergiana* grown in Qatar.

Bacterial Species Tested	Extract *	% Control at 200 µg/mL **
*E. coli*	Acetone	85
	Butanol	0
	Ethanol	85
	Ethyl acetate	55
	Methanol	10
*Bacillus* sp.	Acetone	65
	Butanol	17
	Ethanol	11
	Ethyl acetate	15
	Methanol	0
*Staphylococcus aureus*	Acetone	40
	Butanol	65
	Ethanol	22
	Ethyl acetate	33
	Methanol	45

* The extracts were prepared as described in materials and methods. **Ampicillin (100 µg/mL) was used as positive control.

**Table 3 molecules-27-06400-t003:** The antibacterial activities of various extracts of *Acacia*
*ehrenbergiana* growing in the desert environment: A comparison.

Plant Source	Extract	Organism			Part of Plant Tested	Ref
		*E. coli*	*Bacillus*	*S. aureus*		
Qatar	Acetone	+	+	+	Stem, leaves Concentrations used: 5–200 µg/mL	Present Study
	Ethanol	+	+	+		
	Butanol	−	+	+		
	Methanol	+	−	+		
Sudanese Marawi north desert	Methanol	+	+	+	Stem bark Concentration used: 200 mg/mL	[12]
Egypt, Aswan	Ethanol	+	+	+	Extracted compounds from aerial parts	[14]
Saudi Arabia	Methanol ^1^	+	+	+	Fresh aerial parts, leaves	[4,8] [16,22]
	Ethanol	+	ND	+		

ND = Not determined. ^1^ At 10 mg/mL, weak activity was observed against the tested organisms in contrast to high activity against *K. pneumonia*.

**Table 4 molecules-27-06400-t004:** The antibacterial activity of the metabolites identified in the acetone extracts of *Acacia ehrenbergiana*.

Isolated Compound	Organism	Results *	Literature Report *
Stigmasterol	*E. coli*	Inactive	Active [23]
Spinasterol		Inactive	N/A
Theogallin		Inactive	N/A
Stigmasterol	*S. aureus*	Inactive	Active [23]
Spinasterol		Inactive	N/A
Theogallin		Inactive	N/A

* Reference value: Active ≡ M.D.I.Z ≥ 14 mm, moderately active ≡ M.D.I.Z. = 10–13 mm, Inactive ≡ M.D.I.Z. < 10 mm [19]. N/A = Not available.

**Table 5 molecules-27-06400-t005:** The antifungal activity of the metabolites identified in the acetone extracts of *Acacia ehrenbergiana*.

Isolated Compound	Organism	Results *	Literature Report *
Stigmasterol	*Saccharomyces cerevisiae*	Inactive	Active [23]
Spinasterol		Inactive	N/A
Theogallin		Inactive	N/A
Stigmasterol	*A. fumigatus*	Inactive	Active * [23]
Spinasterol		Inactive	N/A
Theogallin		Inactive	N/A

* Reference value: Active ≡ M.D.I.Z ≥ 14 mm, moderately active ≡ M.D.I.Z. = 10–13 mm, Inactive ≡ M.D.I.Z. < 10 mm [19].

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
