# Peer review of "Phytochemical Analysis of Acaciaehrenbergiana (Hayne) Grown in Qatar: Identification of Active Ingredients and Their Biological Activities"

_molecules, 2022, doi:10.3390/molecules27196400_

Round 1

Reviewer 1 Report

Major revision required:

The article entitled: ‘Phytochemical Analysis of Acacia ehrenbergiana (Hayne) 2 Grown in Qatar: Identification of Active Ingredients and Their Biological Activities’ by Vandana Thotathil, Hanan Rizk, Ameena Fakhroo and Lakshmaiah Sreerama lacks novelty since it is reporting the phytoconstituents of a desert plant which has already been studied extensively.

On the above, the article is liable for rejection. However, the article has some publishable content. Hence, the authors may be asked to rewrite it completely to genuinely make it worthy of publishing. (lines: 217-220)

Minor corrections needed:

1.      The authors have to improve their writing. See line 221: ‘….. and Staphylococcus aureus were most the susceptible 221 organisms to methanol …’. This is awkward.

2.      The terms, antibacterials – antimicrobials are used inconsistently; for the same meaning or different meaning

3.      Different spectroscopic/spectrometric methods have been used. But the results are not linked to the data acquired by such methods. These may be deposited as supplementary materials for future use by others

4.      Table 3 has a problem. The plant collected from Sudanese Marawi north desert was extracted with chloroform and compared with the other collections. Report only comparable protocols

5. Lines 268-269 report, ‘The anti-inflammatory effects of A. ehrenbergiana extracts could 268 be in part be attributed to stigmasterol and spinasterol compounds.’ Where is the report? Major revision required:

The article entitled: ‘Phytochemical Analysis of Acacia ehrenbergiana (Hayne) 2 Grown in Qatar: Identification of Active Ingredients and Their Biological Activities’ by Vandana Thotathil, Hanan Rizk, Ameena Fakhroo, and Lakshmaiah Sreerama lacks novelty since it is reporting the phytoconstituents of a desert plant which has already been studied extensively.

On the above, the article is liable for rejection. However, the report has some publishable content. Hence, the authors may be asked to rewrite it completely to make it worthy of publishing genuinely. (lines: 217-220)

Minor corrections needed:

1.      The authors have to improve their writing. See the line 221: ‘….. and Staphylococcus aureus were most the susceptible 221 organisms to methanol …’. This is awkward.

2.      The terms, antibacterials – antimicrobials are used inconsistently; for the same meaning or different meaning

3.      Different spectroscopic/spectrometric methods have been used. But the results are not linked to the data acquired by such methods. These may be deposited as supplementary materials for future use by others

4.      Table 3 has a problem. The plant collected from Sudanese Marawi north desert was extracted with chloroform and compared with the other collections. Report only comparable protocols

5.      The lines 268-269 report, ‘The anti-inflammatory effects of A. ehrenbergiana extracts could 268 be in part be attributed to stigmasterol and spinasterol compounds.’ Where is the report?

Author Response

Major revision required:

(* the edited paper is attached after the report)

The article entitled: ‘Phytochemical Analysis of Acacia ehrenbergiana (Hayne) Grown in Qatar: Identification of Active Ingredients and Their Biological Activities’ by Vandana Thotathil, Hanan Rizk, Ameena Fakhroo and Lakshmaiah Sreerama lacks novelty since it is reporting the phytoconstituents of a desert plant which has already been studied extensively.

On the above, the article is liable for rejection. However, the article has some publishable content. Hence, the authors may be asked to rewrite it completely to genuinely make it worthy of publishing. (lines: 217-220)

Response: Thank you for the comment and it is fair.  It is true that this plant has been studied (reference 4, 12 and 25) however the compounds being reported is new.  Papers representing the previous work, we believe, are adequately cited with most appropriate references. Taking into consideration the comments of the reviewer, we have expanded introduction and added additional discussion as well as new references (2,15,16), especially as related to lines 217-220 indicated by the reviewer.  Further, the paper has been thoroughly reviewed and rewritten as per the suggestions of the reviewer.

Minor corrections needed:

  1. The authors have to improve their writing. See line 221: ‘….. and Staphylococcus aureus were most the susceptible organisms to methanol …’. This is awkward.

Response: We have revised this sentence and reworded it.

  1. The terms, antibacterials – antimicrobials are used inconsistently; for the same meaning or different meaning

Response: All of them are now changed to “antibacterial” to be consistent throughout the manuscript.

  1. Different spectroscopic/spectrometric methods have been used. But the results are not linked to the data acquired by such methods. These may be deposited as supplementary materials for future use by others.

Response: Thank you for the suggestion, spectroscopic data is now included as the supplementary file. If necessary we are willing to include them in the manuscript, however, the manuscript will become too long. The methods section related to this section is now expanded and improved.

  1. Table 3 has a problem. The plant collected from Sudanese Marawi north desert was extracted with chloroform and compared with the other collections. Report only comparable protocols.

Response: Table 3 has been corrected as suggested by the reviewer. Now the comparison are against the solvents we have used for extraction of phytochemicals.

  1. Lines 268-269 report, ‘The anti-inflammatory effects of A. ehrenbergiana extracts could be in part be attributed to stigmasterol and spinasterol compounds.’ Where is the report? 

Response: References (29, 30) have been cited to address this point as well as the sentences have been rewritten.

Reviewer 2 Report

The authors should consider other journals to publish this manuscript. This manuscript is not compatible with this journal. 

There are some typos. For example,

1. anthroquinone -> anthraquinone

2. pre dominant -> predominant

3. phlobatatannins -> phlobatannins

4. anthrocyanins -> anthocyanins

5. E .coli -> E. coli

6. could be in part be attributed to -> could be in part attributed to (line 268)

Some inappropriate references

reference 14: no metric data for this reference journal. Change to a more relevant journal.

The discrepancy between antibacterial/antifungal activity in acetone extracts and literature value is difficult to understand. 

Author Response

Reviewer 2:

(* the edited paper is attached after the report for reference)

The authors should consider other journals to publish this manuscript. This manuscript is not compatible with this journal.

Response: The work is more relevant to the journal of Molecules and therefore it was submitted here. Molecules has published this kind of work extensively and this has been out motivation to submit it here.

There are some typos. For example,

  1. anthroquinone -> anthraquinone
  2. pre dominant -> predominant
  3. phlobatatannins -> phlobatannins
  4. anthrocyanins -> anthocyanins
  5. E .coli -> E. coli
  6. could be in part be attributed to -> could be in part attributed to (line 268)

Response: Items 1-6 above have been corrected. Thank you.

Some inappropriate references

reference 14: no metric data for this reference journal. Change to a more relevant journal.

Response: In addition to reference 14( new number18), we have added the original reference from where the methods were implemented (19,20). Reference 14 ( new number18)has the details of any modifications that were adopted in our lab, therefore we like to retain this reference.

The discrepancy between antibacterial/antifungal activity in acetone extracts and literature value is difficult to understand.

Response: The table has been corrected and simplified.

Reviewer 3 Report

The manuscript presents a diverse study to frame a paper. Why different extracts were used is not clear, to me to generate data that are not useful. It is basically an elementary study of a desert plant without any proper objective and design. There is the claim of identification of compounds by IR, NMR etc but no details of methods or data are given. How structures were derived? Not suitable for publication in its present form.

Author Response

Reviewer 3

(* the edited paper is attached after the report for reference)

The manuscript presents a diverse study to frame a paper. Why different extracts were used is not clear, to me to generate data that are not useful.

Response: It is known that phytochemicals are selectively extracted into different organic solvents. With this in mind, we have attempted to see which solvents are more effective.  Our ultimate goal is to analyze all these fractions and identify the active compounds.  This part of the work is in progress. As part of this paper, we have analyzed Acetone extracts in detail.

It is basically an elementary study of a desert plant without any proper objective and design.

Response: The qualitative work is of preliminary nature, no doubt.  However, we have done a detailed analysis and identified 3 new compounds in the acetone extracts that has not been reported previously.

There is the claim of identification of compounds by IR, NMR etc but no details of methods or data are given. How structures were derived?

Response: The methods are now rewritten and details are added.  The data related to these studies is being submitted as Supplementary file.  If necessary, we are willing to include this data as part of the manuscript.

Not suitable for publication in its present form.

Response: The manuscript has been extensively rewritten as the reviewer’s suggested.

Round 2

Reviewer 3 Report

The manuscript has been thoroughly revised and the authors have replied to the satisfaction. Therefore, the manuscript is acceptable.

Author Response

Dear Dr. Supreeya Srisuk

Editor, Molecules

Re:     Manuscript - Molecules-1905887 (Phytochemical Analysis of Acacia ehrenbergiana (Hayne) Grown in Qatar: Identification of Active Ingredients and Their Biological Activities) by Vandana Thotathil, Hanan Rizk, Ameena Fakhroo and Lakshmaiah Sreerama.

We would like to appreciate your valuable time and efforts for reviewing our paper, and providing comments for the betterment and improvement of the article. We have carefully considered each of your comments and addressed them to the best of our abilities. Mentioned below are the modifications done based on the reviewers’ comments. All the changes made are marked up using “Track Changes”

  • The manuscripts has been thorough “Spell Checked” and added minor edits to improve the quality of the manuscript further.
  • All references have been revised, formatted where necessary and checked their authenticity
  • English language revised to meet the standards of the journal.

Yours Sincerely,

Prof Lakshmaiah Sreerama

Professor of Biochemistry

College of Arts and Science

Department of Chemistry and Earth Sciences Qatar University, P.O. Box 2713, Doha, Qatar Fulbright Fellow 2010-11, Tribhuvan University, Kathmandu, Nepal

e-mail: [email protected]   |  Tel: (+974) 4403 6542 | Mob. (+974) 7070 6746